# Improved Electron Efficiency of Zero-Valent Iron towards Cr(VI) Reduction after Sequestering in Al_2_O_3_ Microspheres

**DOI:** 10.3390/ijerph19148367

**Published:** 2022-07-08

**Authors:** Chuan Wang, Sha Wang, Cheng Song, Hong Liu, Jingxin Yang

**Affiliations:** 1Key Laboratory for Water Quality and Conservation of the Pearl River Delta, Ministry of Education, Institute of Environmental Research at Greater Bay, Guangzhou University, Guangzhou 510006, China; wangchuan@gzhu.edu.cn; 2Chongqing Institute of Green and Intelligent Technology, Chinese Academy of Sciences, Chongqing 400714, China; wangsha@cigit.ac.cn (S.W.); songcheng@cigit.ac.cn (C.S.); liuhong@cigit.ac.cn (H.L.); 3Key Laboratory of Reservoir Aquatic Environment, Chinese Academy of Sciences, Chongqing 400714, China

**Keywords:** electron efficiency, zero-valent iron, ZVI sequestered Al_2_O_3_ composite

## Abstract

Zero-valent iron (ZVI) is widely used for groundwater remediation, but suffers from high electron consumption because of its free contact with non-target substrates such as O_2_. Here, ZVI-ALOX particles were prepared via in situ NaBH_4_ aqueous-phase reduction of ferrous ions (Fe^2+^) preabsorbed into Al_2_O_3_ microspheres. The electron efficiency (EE) and long-term performance of the material were improved by sequestering ZVI in the interspace of the Al_2_O_3_ microspheres (ZVI-ALOX). During long-term (350 days) continuous flow, Cr(VI) was removed to below the detection limit for over 23 days. Based on the high reactivity of ZVI towards Cr(VI), the EE of ZVI-ALOX was evaluated by measuring its Cr(VI) removal efficiency at neutral pH and comparing it with that of ZVI. The results showed that the EE of ZVI-ALOX during long-term continuous flow could reach 39.1%, which was much higher than that of ZVI (8.68%). The long-term continuous flow results also demonstrated that treatment of the influent solution achieved higher EE values than in the batch mode, where the presence of dissolved oxygen reduced EE values. At lower pollutant concentrations, the sequestering of ZVI was beneficial to its performance and long-term utility. In addition, measurement of the acute toxicity of treated column effluent using the indicator organism *Photobacterium phosphoreum* T_3_ showed that ZVI-ALOX could reduce the toxicity of 5 mg/L Cr(VI) solution by ~70% in 350 d. The results from this study provide a basis for the development of permeable reactive barriers for groundwater remediation based on sequestered ZVI.

## 1. Introduction

Zero-valent iron (ZVI) has been successfully used in recent years for the remediation of groundwater contaminated with a wide range of organic and inorganic pollutants [1,2,3]. The standard redox potential (E^0^) for ZVI (Fe^2+^/Fe^0^) of −0.44 V is much lower than that of many contaminants [4]. A great deal of research has focused on the removal of contaminants by ZVI, because of its low toxicity, abundance, ease of production, and simple use [5]. However, due to high surface reactivity and magnetic properties, ZVI particles are susceptible to agglomeration and oxidation (surface passivation) following contact with air or water, leading to the loss of reactivity. The usual practice has been to use excess ZVI materials to compensate for the waste of electrons, or to improve the activity of ZVI materials to increase the rate of pollutant removal, while ignoring the economic importance of chemical consumption efficiency [6]. Therefore, it is necessary to investigate the electron efficiency (EE) of ZVI materials for evaluating their practical engineering application potential.

One of the practical limitations of ZVI is its undesired electron consumption, due to contact with non-target substrates [7,8]. For example, the autocatalytic anaerobic corrosion of ZVI occurs in deep groundwater systems according to Equation (1): (1)$$\mathrm{Fe}+H_{2}O\to {\mathrm{Fe}}^{2+}+2{\mathrm{OH}}^{-}+H_{2}$$

Similarly, low concentrations of dissolved O_2_ present in shallow groundwater can accelerate the electron consumption of ZVI via Equation (2):(2)$$\mathrm{Fe}+O_{2}+2H_{2}O\to 4{\mathrm{OH}}^{-}+2{\mathrm{Fe}}^{2+}$$

Furthermore, these issues are compounded by the excess quantities of non-target oxidants (especially water) relative to the dilute concentrations of target pollutants [9,10]. To compensate for this undesired electron consumption, the practical application of ZVI requires overdosing, which can result in residual material following groundwater remediation [7,8]. One method to reduce undesired electron consumption and improve electron transfer to target substrates is selectively to prevent contact between non-target substrates and ZVI. While various coatings have been applied to ZVI to prevent aggregation and increase mobility [11,12,13], their effectiveness towards the inhibition of corrosion reactions has not yet been documented. 

Chromium (Cr), one of the alarming metallic pollutants, can exist in soil and groundwater [14]. Although Cr(III) is an essential micronutrient in metabolic activity, Cr(VI) poses a greater threat due to its mobility, bioavailability, carcinogenicity, and mutagenicity [15,16]. Therefore, toxic water soluble contaminants Cr(VI) (electron acceptor) can be reduced by ZVI (electron donor) to less-toxic species such as Cr(III) [17].
(3)$${\mathrm{Cr}}_{2}O_{4}^{2-}+{\mathrm{Fe}}^{0}+4H_{2}O\to {\mathrm{Cr}}^{3+}+{\mathrm{Fe}}^{3+}+8{\mathrm{OH}}^{-}$$
(4)$$2{\mathrm{Cr}}_{2}O_{4}^{2-}+3{\;\mathrm{Fe}}^{0}+16H^{+}\to 2{\mathrm{Cr}}^{3+}+3{\mathrm{Fe}}^{2+}+8H_{2}O$$
(5)$${\mathrm{Cr}}_{2}O_{4}^{2-}+3{\mathrm{Fe}}^{2+}+8H^{+}\to {\mathrm{Cr}}^{3+}+3{\mathrm{Fe}}^{3+}+4H_{2}O$$

Certain ZVI based composites have been used to remove Cr(VI) from wastewater, such as biochar-supported ZVI [18,19], clay ZVI composites [20], silica-supported bimetallic ZVI [21,22], as well as combined effects of metal doping and polymer coating on ZVI materials [23]. However, these studies mainly examined the removal kinetics of chromium, its migration properties, etc., and rarely studied the EE of these materials. However, quantification of the EE provides a basis to improve the economic efficiency of materials. Thus, the EE factor should be intelligently balanced against the reaction rate to assess ZVI-based groundwater remediation and wastewater treatment [6].

The aim of this study was to increase the selective electron transfer of ZVI and hence reduce electron wastage by sequestering it from water and oxygen in Al_2_O_3_ microspheres. Al_2_O_3_ microspheres have been used extensively as support materials [24] because of their porosity, large specific surface area, high compressive strength, and resistance to acid and alkali. Fe^2+^ ions, preabsorbed into Al_2_O_3_ microspheres, were reduced in situ with NaBH_4_ to form sequestered ZVI (ZVI-ALOX). The performance of ZVI-ALOX towards the reduction of Cr(VI) was evaluated at neutral pH, and its electron efficiency was evaluated in batch and long-term column flow removal experiments. The findings in this study are expected to lay a foundation for the development of a novel reactive material employing permeable reactive barrier (PRB) technology for the remediation of polluted groundwater.

## 2. Materials and Methods

### 2.1. Materials and Chemicals

Al_2_O_3_ microspheres were obtained from Pingxiang Global Chemical Packing Co., Ltd., (Pingxiang, China). Ferrous sulfate hepta-hydrate (FeSO_4_·7H_2_O), sodium borohydride (NaBH_4_) and potassium dichromate (K_2_Cr_2_O_7_) were purchased from Kelong Chemical Reagent Co., Ltd., (Chengdu, China). All other chemicals were commercially available analytical grade reagents, used without further purification. Deionized water (18.2 MΩ·cm) was used in all experiments. 

### 2.2. Preparation of ZVI and ZVI-ALOX

FeSO_4_·7H_2_O (5.56 g) was dissolved in oxygen-free deionized (100 mL) water at pH 2 under N_2_ atmosphere. Al_2_O_3_ microspheres were calcined at 400 °C for 2 h, then added to the FeSO_4_ solution and stirred for 6 h under N_2_ atmosphere (to prevent the oxidization of Fe^2+^). Subsequently, NaBH_4_ (1.0 g) was added to the mixture over a period of 8 h. The resultant suspension was filtered to remove the microspheres, which were washed three times with oxygen-free deionized water, 1:1 ethanol, and pure ethanol, respectively. The final product was dried overnight at 60 °C under vacuum.

Control material (ZVI, without Al_2_O_3_ microspheres) was prepared in the same way by adding NaBH_4_ to the aqueous FeSO_4_·7H_2_O solution.

### 2.3. Characterizations and Measurements

Scanning electron microscopy (SEM) was performed using a JSM-7800F (JEOL, Japan); images were obtained at an operating voltage of 5.0 kV. Specific surface areas were obtained by the Brunauer–Emmett–Teller (BET) method from the corresponding N_2_ adsorption isotherms using a Belsorp-max surface area and porosity analyzer (MicrotracBEL, Japan). The phase structures were determined by X-ray diffraction (XRD, PANalytical-PW3040/60, The Netherlands). The XPS spectra were recorded on an ESCALAB 250 photoelectron spectrometer (ThermoVG Scientific, USA) with Al Kα (1486.6 eV) as the X-ray source. Cr(VI) was determined colorimetrically by the 1,5-diphenylcarbazide method; absorbance at 540 nm was monitored with a TU-1901 UV–Vis spectrophotometer (Persee Analytics, Inc., Beijing, China).

### 2.4. Batch Experiments

Batch experiments were performed in a 150 mL stoppered conical flask. The initial concentrations of Cr(VI) were 5.0, 2.0, 1.0, and 0.5 mg/L. Aliquots of Cr(VI) solution (100 mL) were treated with ZVI or ZVI-ALOX at a Fe^0^/Cr(VI) ratio of 100 *w*/*w*. H_2_SO_4_ (0.1 M) and NaOH (0.1 M) were used to adjust the pH to 7. The removal efficiencies of each ZVI were calculated from measurements of the Cr(VI) concentrations, before and after the addition of ZVI/ZVI-ALOX. When the concentration of Cr(VI) in the supernatant decreased to below the limit of detection, the Cr(VI) solution was discarded (one sequence) and a further 100 mL Cr(VI) solution at the same initial concentration was added to start the next cycle. This sequence was repeated at each of the four concentration levels until the performance of the ZVI deteriorated significantly. All experiments were performed in duplicate and the results were recorded as mean values.

### 2.5. Column Flow Experiments

Plexiglass tubes (45.00 cm long × 2.35 cm i.d.) were packed with ZVI-ALOX (53 g) or ZVI (mixed with quartz sand and glass wool to the same iron content as ZVI-ALOX) to give a bed depth of 11.50 cm; a 20-mesh stainless steel frit was placed at the bottom of the column to support the packing material. The flow of Cr(VI) solution (5 mg/L, pH 7) through the fixed-bed of ZVI/ZVI-ALOX under gravity was controlled by a valve at the top of the column; the supply of Cr(VI) solution was maintained by pumping the solution from a reservoir at 0.18 mL/min. The concentration of Cr(VI) measured in the effluent was used to determine the removal efficiencies of Cr(VI) by ZVI/ZVI-ALOX.

### 2.6. Luminous Bacteria Toxicity Assay

Changes in the toxicity of the Cr(VI) solution during the reaction with ZVI/ZVI-ALOX were assessed by measuring the reduction in light emission by *Photobacterium phosphoreum* T_3_ under toxic stress. All bioassays were carried out in triplicate and light output was measured using a DXY-2 portable luminometer; *P. phosphoreum* T_3_ and the luminometer were obtained from the Institute of Soil Science, Chinese Academy of Sciences (Nanjing, China). The inhibition ratio was calculated by Equation (6):(6)$$\mathrm{Inhibition}\;\mathrm{ratio}=\frac{I_{0}-I}{I_{0}}\times 100\%$$
where *I*_0_ is the luminescent intensity of the control experiment, and *I* is the luminescent intensity of the sample.

### 2.7. Electron Efficiency Measurements

The electron efficiency (EE) can be defined as the percentage of electrons utilized via substrate reduction (N_e_) over an entire consumption (N_t_) within a specific time interval [25]. Hence, EE can be used to quantify efficiency of the target reduction, according to Equation (7):(7)$$\mathrm{EE}=\frac{N_{e}}{N_{t}}\times 100\%$$

Equations (3)–(5) show that Fe^0^ can oxidize Cr(VI) to give Cr(III), and that the Fe^II^ formed can also reduce Cr(VI) to Cr(III) as it transforms into Fe^III^. Therefore, one mole of Fe^0^ consumes three mole of electrons, while N_t_ is related the amount of Fe^3+^. Values of N_e_ and N_t_ can be calculated from Equations (8) and (9):(8)$$N_{e}=3M_{\mathrm{Cr}\left(\mathrm{III}\right)}$$
(9)$$N_{t}=3M_{\mathrm{Fe}\left(\mathrm{III}\right)}$$
where M_Cr(III)_ and M_Fe(III)_ are the amounts of Cr(III) and Fe(III) (mol). 

EE measurements were made in the batch and column experiments using ZVI and ZVI-ALOX. The concentrations of Cr(VI) in solution with ZVI, and those of Cr(VI), Fe^0^ and total Fe using ZVI-ALOX, were measured before and after the reaction. The decrease of Cr(VI) in solution was related to the production of Cr(III) and the adsorption of Cr(VI) by ZVI-ALOX. Hence, the amount of Cr(III) product could be obtained by the differences between the initial Cr(VI) concentration and those in the solution containing ZVI-ALOX. Because Fe^0^ transforms into Fe(II) in an acid condition, ZVI-ALOX was ground and dissolved in 1 M HCl to quantify the amounts of Fe^0^ and unreacted Fe^0^ via the measurement of Fe(II). This could then be used to determine the amount of Fe(III) formed.

## 3. Results and Discussion

### 3.1. Characterization

Morphologies of the Al_2_O_3_ microspheres and ZVI-ALOX are shown in Figure 1. The internal structure of the Al_2_O_3_ microspheres was porous and rough, which was beneficial for sequestering the ZVI (Figure 1a). The ZVI particles were spherical or near-spherical in shape (80–100 nm), and located in the pores or channels of the microspheres (Figure 1b). The elemental composition and calculated mass ratio of Fe (5.92 wt%) obtained by EDS indicated that ZVI was successfully incorporated within the Al_2_O_3_ microspheres. At the same time, diffraction peaks corresponding to Al_2_O_3_ and the ZVI were observed in the XRD spectra of ZVI-ALOX (Figure 2).

The specific surface areas and pore volumes of the Al_2_O_3_ microspheres and ZVI-ALOX, before and after Cr(VI) reduction, are given in Table 1. The largest and smallest specific surface areas were observed in the Al_2_O_3_ microspheres and ZVI-ALOX, respectively. The decrease of both features in ZVI-ALOX was due to the restriction of the pores and channels of Al_2_O_3_ by the ZVI particles, which agreed with the SEM results. The increase in the surface area and pore volume after Cr(VI) reduction suggests that some ZVI was lost by dissolution of the iron products.

### 3.2. Batch Experiments

The sequential batch experiments comparing the removal efficiencies of Cr(VI) using ZVI-ALOX and ZVI over the initial concentration range 0.5–5.0 mg/L are shown in Figure 3. Notably, ZVI-ALOX maintained better continuous Cr(VI) removal performance compared with ZVI: at 5.0 mg/L Cr(VI) solution, the removal efficiency of ZVI-ALOX was 100% after 7 d, and this performance was maintained for further three cycles (i.e., a total of 28 d). In comparison, the first cycle with ZVI at 5.0 mg/L Cr(VI) achieved 100% removal in 4 d, the second cycle required 14 d to achieve 100% removal, and removal ability decreased greatly thereafter. Similar trends were observed at lower initial concentrations of Cr(VI). Compared with ZVI, which was in direct contact with the Cr(VI) solution, the Al_2_O_3_ microspheres may increase the mass transfer resistance of Cr(VI), thereby delaying its contact with the sequestered ZVI. Hence, the initial removal of Cr(VI) by ZVI was relatively rapid. However, subsequent oxidation when exposed to air or water rapidly reduced its efficiency (Figure 3b). On the other hand, the Al_2_O_3_ matrix surrounding ZVI may reduce contact with the non-target substrates such as dissolve oxygen (DO) and water, enabling optimum removal performance over a greater time period.

### 3.3. Column Flow Experiments and Toxicity Studies

Column studies were used to evaluate the long-term Cr(VI) removal performance of ZVI-ALOX;the results obtained at neutral pH, and the comparison with ZVI, are given in Figure 4a. For ZVI-ALOX, the concentration of Cr (VI) in the effluent remained below the detection limit (i.e., 100% removal) during the first 23 days, decreasing thereafter until loss of performance at 350 days. A total of 92 L Cr(VI) solution passed through the fixed-bed column, from which 306.36 mg of Cr(VI) was removed. The equivalent experiment using ZVI showed that complete removal of Cr(VI) from the influent was maintained for 2 d, and thereafter the concentration of Cr(VI) in the effluent increased quickly, approaching the initial influent Cr(VI) concentration from day 30 onwards. The volume of feed solution was 27 L and the total removal amount of Cr(VI) was 33.48 mg. The long-term reactivity of ZVI-ALOX could also be attributed to the Al_2_O_3_ microsphere matrix, which may have shielded the ZVI from non-target oxidants.

The toxicity of the effluent from column experiments using ZVI and ZVI-ALOX was evaluated using the luminescent bacteria (*P. phosphoreum* T_3_) test [25]. Bioluminescence intensity was inversely proportional to the toxicity of the solution, and the inhibition rates were calculated from Equation (6). As shown in Figure 4b, inhibition of *P.* phosphoreum by Cr(VI) in the effluents from the ZVI and ZVI-ALOX treatments of Cr(VI) solution (5 mg/L) decreased rapidly over the first 12 d, reaching minimum values of ~8% and ~30% respectively. Beyond this period, the toxicity of effluent from ZVI-ALOX remained relatively low (inhibition rate < 20%) while that of ZVI increased rapidly.

### 3.4. Transformation of Cr and Fe Species in Reactions of Cr(VI) with ZVI-ALOX

The XPS spectra of ZVI-ALOX before the Cr(VI) removal reaction is shown in Figure 5. Transformation of Cr(VI) into Cr(III) in the column experiments using ZVI-ALOX was demonstrated by XPS analysis of the solid material recovered after the removal reaction (Figure 6). The XPS wide scan survey spectrum of ZVI-ALOX after Cr(VI) removal (Figure 6a) revealed new peaks at 580.0 eV, indicating the absorption of chromium species on the ZVI-ALOX particle surface [26]. The high resolution XPS spectrum of the Cr 2p region (Figure 6b) showed Cr 2p1/2 and Cr 2p3/2 peaks at 586.9 and 577.2 eV, respectively. The broad peak of Cr 2p3/2 could be fitted to two peaks at binding energies of 578.1 and 576.3 eV, which were characteristic of Cr(VI) and Cr(III), respectively [27]. This suggested that both Cr(VI) and Cr(III) coexisted on the surface of Cr(VI)-adsorbed ZVI-ALOX. Hence, ZVI-ALOX favored the reduction of Cr(VI) to Cr(III). The high resolution Fe2p spectra (Figure 6c) of Fe 2p1/2 (725.0 eV) and Fe 2p3/2 (711.0 eV) corresponded to oxidized iron species. Deconvolution of Fe 2p3/2 into two peaks at 710.7 eV and 713.5 eV indicated that Fe^3+^ was the major component of the iron oxides, in the form of Fe_2_O_3_ and a mixed phase of Cr_x_Fe_1−x_(OH)_3_ [17]. The small peak at 706.0 eV due to Fe^0^ species [28] agreed with the exposure of fresh ZVI-ALOX. As shown in Figure 6d, the O1s binding energies at 530.5 eV and 531.7 eV corresponded to the presence of O^2–^ and OH^–^ groups, respectively. The shift of each oxygen peak by ~0.5 eV suggested that oxygen active groups were involved in the redox reaction between Fe and Cr [29]. 

### 3.5. EE Measurements

EE is a quantitative measure of electron utilization during the ZVI target reduction process, which provides a basis to improve economic efficiency. The physical structures of ZVI, initial concentration of the target solution, mode of reaction and the substrate system all affected EE values (Figure 7). Figure 7a shows that the EE of ZVI-ALOX and ZVI both increased with increasing initial Cr(VI) concentration. Compared with ZVI, the values of EE obtained with ZVI-ALOX were higher at all concentrations, especially at 5 mg/L Cr(VI). This suggested that the interface between Al_2_O_3_ and ZVI in the microspheres played an important role in the utilization of ZVI. Both ZVI-ALOX and ZVI gave higher EE values in the column removal mode. Obviously, compared with the batch removal of Cr (VI), the EE values of isolated ZVI were significantly higher (39.1% vs. 5.12%) (Figure 7b). 

## 4. Conclusions

In order to reduce unnecessary electronic consumption of ZVI materials and to improve material utilization, this study prepared ZVI-ALOX for the removal of chromium from groundwater or wastewater. ZVI-ALOX was prepared by using ferrous ions pre-adsorbed in Al_2_O_3_ microspheres with NaBH_4_ in situ reduction, and ZVI particles were dispersed inside the Al_2_O_3_ microspheres. Compared with ZVI, the sequestered ZVI demonstrated significant long-term activity towards Cr(VI) reduction. The EE of ZVI-ALOX was higher than that of ZVI, while DO in the influent significantly reduced efficiency due to electron transfer to the non-target substrate. In addition, the treatment of Cr(VI) using the continuous column flow mode gave higher EE values compared with the batch mode. The results from this study provide a basis for the development of ZVI-ALOX in PRB for long-term groundwater remediation. For the practical application of ZVI technology, the EE value should be used as the basis for material selection. Further research may focus on testing in a continuous manner in more realistic systems (e.g., using actual bodies of water, considering ion effects, etc.).

## Figures and Tables

**Figure 1 ijerph-19-08367-f001:**
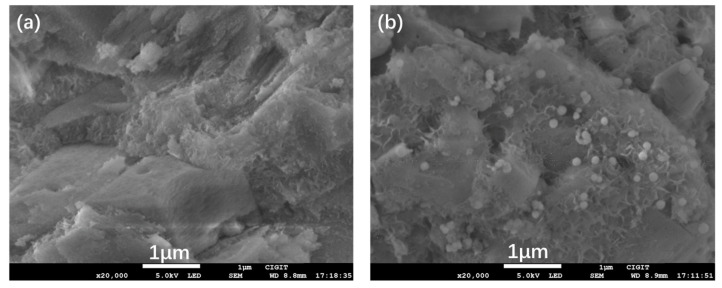
SEM images of the Al_2_O_3_ and ZVI-ALOX particles: (**a**) Al_2_O_3_ microspheres; (**b**) ZVI-ALOX microspheres.

**Figure 2 ijerph-19-08367-f002:**
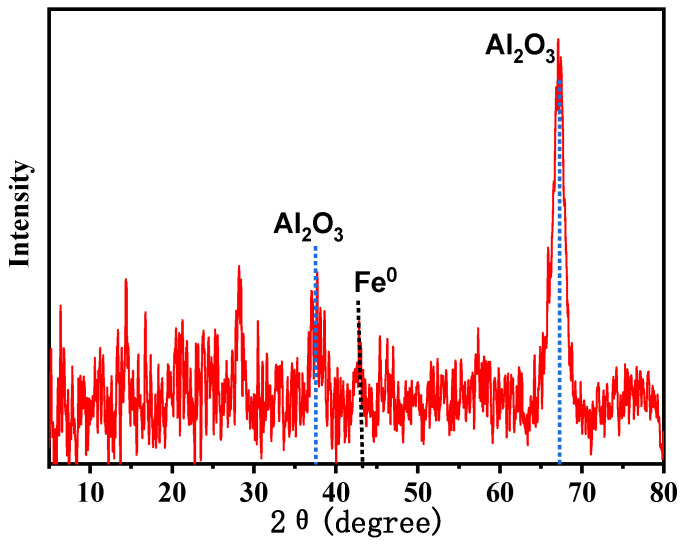
XRD spectra of ZVI-ALOX.

**Figure 3 ijerph-19-08367-f003:**
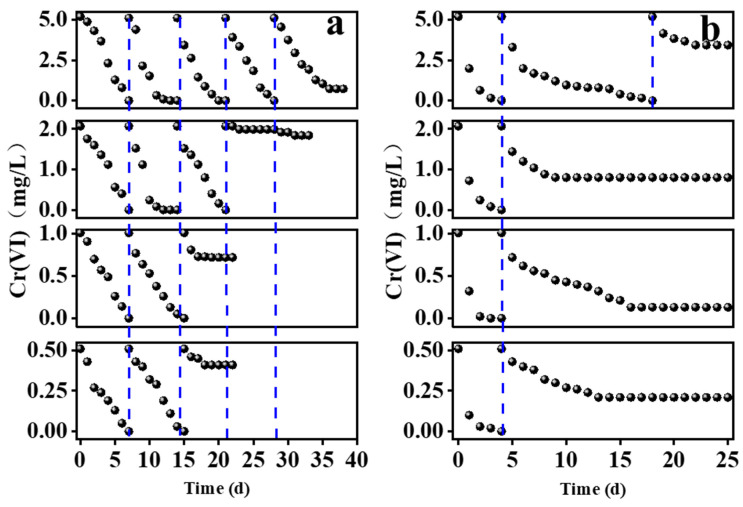
Sequential batch experiments comparing the Cr(VI) removal efficiencies of the different ZVI materials over the initial concentration range 0.5–5.0 mg/L: (**a**) ZVI-ALOX; (**b**) ZVI.

**Figure 4 ijerph-19-08367-f004:**
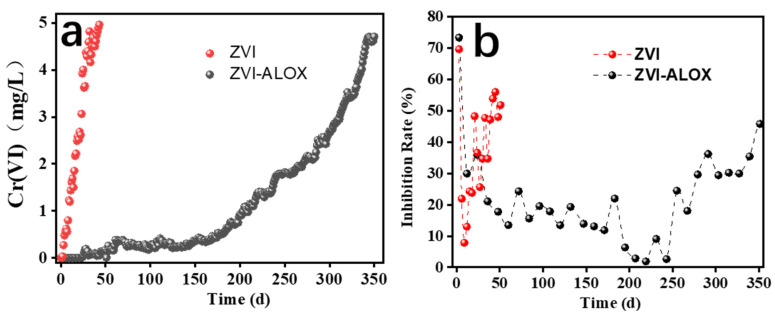
Long term removal experiments using ZVI and ZVI-ALOX (influent [Cr(VI)], 5 mg/L): (**a**) effluent concentrations of Cr(VI); (**b**) inhibition rates of *P. phosphoreum* in the column effluents.

**Figure 5 ijerph-19-08367-f005:**
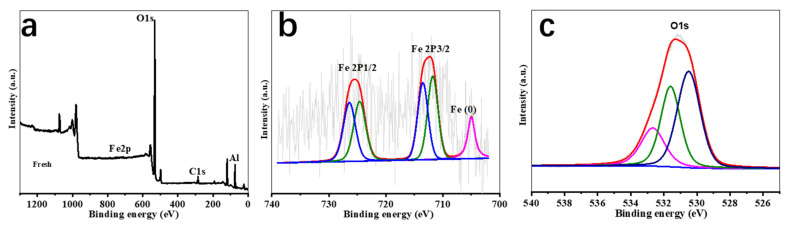
XPS spectra of ZVI-ALOX before the Cr(VI) removal reaction: (**a**) wide scan survey spectrum; (**b**,**c**) high resolution spectra of Fe 2p and O1s.

**Figure 6 ijerph-19-08367-f006:**
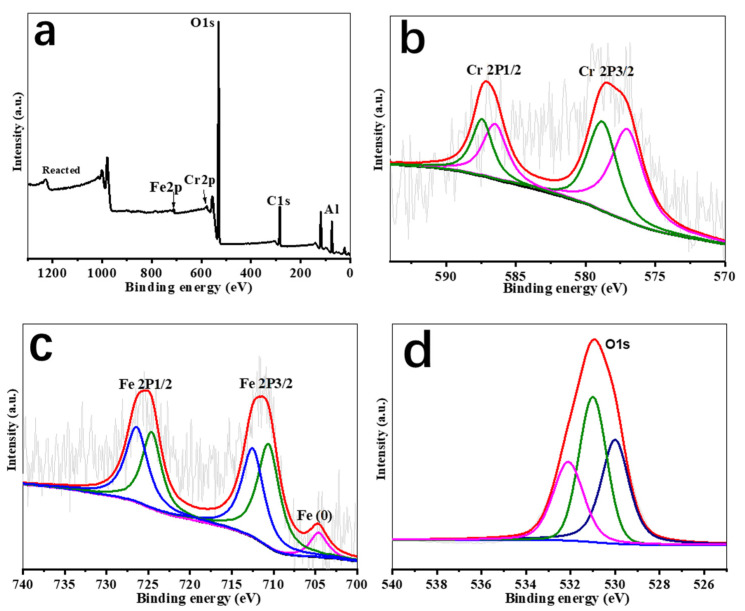
XPS spectra of ZVI-ALOX after the Cr(VI) removal reaction: (**a**) wide scan survey spectrum; (**b**–**d**), high resolution spectra of Cr 2p, Fe 2p and O1s.

**Figure 7 ijerph-19-08367-f007:**
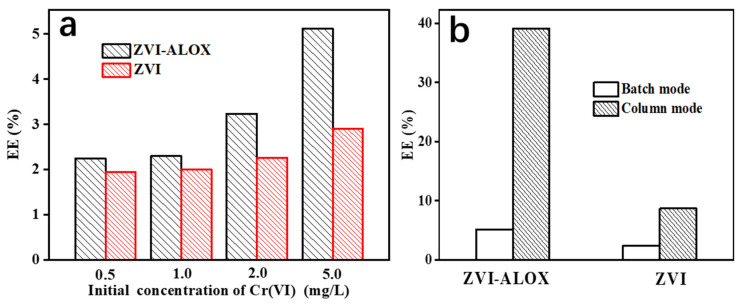
Effects of key parameters on the EE of ZVI-ALOX and ZVI: (**a**) effect of initial Cr(VI) concentration in batch mode; (**b**) effect of batch and column modes (inflow concentration 5 mg/L Cr(VI)).

**Table 1 ijerph-19-08367-t001:** Specific surface areas and total pore volumes (BET method).

Sample	Specific Surface Area (m^2^/g)	V(cm^3^/g)
Al_2_O_3_ microspheres	287.20	0.42
ZVI-ALOX	165.16	0.33
ZVI-ALOX after Cr(VI) reduction (used in continuous tests)	225.47	0.37

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
