# Peer review of "Improved Electron Efficiency of Zero-Valent Iron towards Cr(VI) Reduction after Sequestering in Al2O3 Microspheres"

_ijerph, 2022, doi:10.3390/ijerph19148367_

Round 1
Reviewer 1 Report
This is the review of the manuscript entitled „Improved Electron Efficiency of Zero-valent Iron towards Cr(VI) Reduction after Sequestering in Al2O3 Microspheres”.
The authors present an interesting topic, being in line with the mission of the IJERPH (ISSN 1660-4601) Journal. The writing style is acceptable, and the organization of this manuscript is reasonable.
I have some suggestions:
Introduction section must be written in a more quality way, i.e. more up-to-date references addressed. The novelty of the work must be clearly addressed and discussed, compare your research with existing research findings, and highlight novelty.
- A structure of the manuscript would be welcome, in order to be easier to follow what the authors will present next.
- Also here, the authors could highlight the main strengths of their work.
- The authors can highlight the usefulness of the study in its practical applicability.
- Providing information regarding how the accuracy of the results was verified.
- Conclusion section is missing some perspectives related to the future research work, quantify the main research findings.
Author Response
Thanks for the reviewer’s comments and suggestions. They play an important role in revising our manuscript. Relevant comments and suggestions are modified in the revised manuscript as required.
1. We've re-edited the introduction, added the latest references, and restructured the documentation. The main content of this study is to place ZVI inside the Al2O3 Microspheres to avoid unnecessary electronic consumption, and its electron efficiency is higher than that of ZVI materials. Suitable electron efficiency is the basis for the practical application of ZVI materials, and this study proves through long-term column experiments that ZVI-ALOX has a high electron efficiency and has the potential for groundwater remediation.
2. In the Conclusion section,we’ve added the future research work, quantify the main research findings.” In order to reduce unnecessary electronic consumption of ZVI materials and improve material utilization,this study prepared ZVI-ALOX for the removal of chromium from groundwater or wastewater. ZVI-ALOX is prepared by using ferrous ions pre-adsorbed in Al2O3 microspheres with NaBH4 in situ reduction, and ZVI particles were dispersed inside the Al2O3 microspheres. Compared with ZVI, the sequestered ZVI demonstrated significant long-term activity towards Cr(VI) reduction. The EE of ZVI-ALOX was higher than that of ZVI, while DO in the influent significantly reduced efficiency due to electron transfer to the non-target substrate. In addition, the treatment of Cr(VI) using the continuous column flow mode gave higher EE values compared with the batch mode. The results from this study provide a basis for the development of ZVI-ALOX in PRB for long-term groundwater remediation. Obviously, for the practical application of ZVI technology, the EE value should be used as the basis for material selection. Further research may focus on testing in a continuous manner in more realistic systems (e.g., using actual bodies of water, considering ion effects, etc.).
Reviewer 2 Report
This is a very good paper documenting an interesting and novel method for in-situ water treatment. The only recommendation I have is a minor edit to the abstract. You indicate that over 365 days, ZVI-ALOX shows 39% removal compared to just ZVI. I think this undersells your process. Instead, i would state that Cr(VI) was removed to below the detection limit for over 23 days.
Author Response
Thanks for the reviewer’s comments and suggestions. They play an important role in revising our manuscript. Relevant comments and suggestions are modified in the revised manuscript as required.
Your suggestion on removal efficiency has been added to the abstract. “The electron efficiency (EE) and long-term performance of the material were improved by sequestering ZVI in the interspace of Al2O3 microspheres (ZVI-ALOX). During long-term (350 days) continuous flow Cr(VI) was removed to below the detection limit for over 23 days. Based on the high reactivity of ZVI towards Cr(VI), the EE of ZVI-ALOX was evaluated by measuring its Cr(VI) removal efficiency at neutral pH and comparing it with that of ZVI. The results showed that the EE of ZVI-ALOX during long-term continuous flow could reach 39.1%, which was much higher than that of ZVI (8.68%).”
Reviewer 3 Report
Currently, a lot of research is done on the removal of metals from aqueous solutions (natural waters and wastewater) in the adsorption process. Various adsorbents are used, including waste materials, for example those containing iron and aluminum compounds, such as sludge from the treatment of groundwater and surface waters (water treatment residuals). The authors should place even greater emphasis on the method of obtaining the adsorbent and cost analysis to make the publication more attractive.
The research assumptions are correct and the experiment is done well. It is important that, adsorption was tested for neutral pH, assuming that natural waters will be purified, and that continuous flow laboratory tests were performed, not only batch tests.
The article requires a few editorial corrections. Due to the lack of numbered lines, it is difficult to indicate specific places.
- There is definitely one author (and?) missing.
- A larger font was left for the strain name phosphoreum T3 (abstract, Fig.4), as well as [7].
- Should abbreviations stay in keywords?
- Correct formatting of the references is also required.
- Part of the introduction (last) and formulas 4-9 should, in my opinion, be in the methodology (point 2.7, currently partially not understandable to me).
- The methodology does not list all measurement methods (point 2.3) CrIII, Fe0, XRD, XPS etc.
- Fig.1.b… .showing ZVI in the microspheres - maybe better in micropores
- Tab. 1 - ZVI-ALOX after Cr (VI) reduction - after use in continuous tests?
- 3.2 line 2 concentration range 0.5-5.0 mg/L
- The order of Figs 5 and 6 should be reversed, first before and then after the process
- Bottom p. 8 - although?
- Fig.7. b) …and column mode (inflow concentration 5 mg Cr VI/L)
- Conclusions - Should be improved to emphasize the advantages of using ZVI-ALOX. Currently, unexplained abbreviations DO, PRB appear there.
- …DO in the influent significantly reduced efficiency due to electron transfer ….- was the oxygen concentration or redox in the inflow or during batch tests measured to confirm its effect on a process?
Author Response
Thanks for the reviewer’s comments and suggestions. They play an important role in revising our manuscript. Relevant comments and suggestions are modified in the revised manuscript as required.
We've re-edited the introduction, added the latest references, and restructured the documentation. The main content of this study is to place ZVI inside the Al2O3 Microspheres to avoid unnecessary electronic consumption, and its electron efficiency is higher than that of ZVI materials. Suitable electron efficiency is the basis for the practical application of ZVI materials, and this study proves through long-term column experiments that ZVI-ALOX has a high electron efficiency and has the potential for groundwater remediation.
1 and 2. These document editorial corrections should be the journal compilation conversion mistake. We had line numbers in our original document and didn't have the author (and?) missing or . larger font, we will carefully proofread the manuscript edited by the Journal.
3. It has been deleted.
4. Has been checked and revised.
5. We've re-edited the introduction, added the latest references, and restructured the documentation.
6. Has been checked and added at 2.3 Characterizations and Measurements.
7. Has been checked and revised at line 452-453 “Fig. 1. SEM images of the Al2O3 and ZVI-ALOX particles: (a) Al2O3 microspheres; (b) ZVI-ALOX microspheres”.
8. Has been checked and revised at Tab. 1.
9. Has been checked and revised.
10. Has been reversed.
11. Has been checked and revised.”Obviously, compared with the batch removal of Cr (VI), the EE values of isolated ZVI are significantly higher (39.1%vs 5.12%) (Fig. 7b).’’
12. Has been checked and revised.
13. The influent with N2 bubbling DO ≤0.2 mg/L, and the influent with stirring DO was 4-5 mg/L.
Round 2
Reviewer 1 Report
my suggestions were addressed